# Associations of vaccine status with characteristics and outcomes of hospitalized severe COVID-19 patients in the booster era

Ophir Freund[1]*, Luba Tau[2], Tali Epstein Weiss[1], Lior Zornitzki[1], Shir Frydman[1], Giris Jacob[3], Gil Bornstein[1]

1 Internal Medicine B, Tel-Aviv Sourasky Medical Center and Sackler Faculty of Medicine, Tel Aviv University, Tel Aviv, Israel, 2 Infectious Diseases Unit, Tel-Aviv Sourasky Medical Center and Sackler Faculty of Medicine, Tel-Aviv University, Tel Aviv, Israel, 3 Medicine F, Tel Aviv Medical Center, Sackler Faculty of Medicine, Tel Aviv University, Tel Aviv, Israel

☯ These authors contributed equally to this work.
* ophir068@gmail.com

## Abstract

### Background

The resurgence of COVID-19 cases since June 2021, referred to as the fourth COVID-19 wave, has led to the approval and administration of booster vaccines. Our study aims to identify any associations between vaccine status with the characteristics and outcomes of patients hospitalized with severe COVID-19 disease.

### Methods

We retrospectively reviewed all COVID-19 patients admitted to a large tertiary center between July 25 and October 25, 2021 (fourth wave in Israel). Univariant and multivariant analyses of variables associated with vaccine status were performed.

### Findings

Overall, 349 patients with severe or critical disease were included. Patients were either not vaccinated (58%), had the first two vaccine doses (35%) or had the booster vaccine (7%). Vaccinated patients were significantly older, male predominant, and with a higher number of comorbidities including diabetes, hyperlipidemia, ischemic heart disease, heart failure, immunodeficient state, kidney disease and cognitive decline. Time from the first symptom to hospital admission was longer among non-vaccinated patients (7.2 ± 4.4 days, p = 0.002). Critical disease (p<0.05), admissions to the intensive care unit (p = 0.01) and advanced oxygen support (p = 0.004) were inversely proportional to the number of vaccines given, lowest among the booster vaccine group. Death (20%, p = 0.83) and hospital stay duration (8.05± 8.47, p = 0.19) were similar between the groups.

**Data Availability Statement:** Due to ethical and privacy concerns the primary dataset cannot be made openly available. All other analysis made are given in the Supporting Information files. The study

was done retrospectively and according to the regulations of our institution review board such data could be openly shared. Request for the dataset supporting our results can be made via helsinki@tlvmc.gov.il and will be given by the first author after approval.

**Funding:** The author(s) received no specific funding for this work.

**Competing interests:** The authors have declared that no competing interests exist.

## Conclusion

Hospitalized vaccinated patients with severe COVID-19 had significantly higher rates of most known risk factors for COVID-19 adverse outcomes. Still, all disease outcomes were similar or better compared with the non-vaccinated patients.

## Introduction

Since June 2021, a resurgence of coronavirus 2019 (Covid-19) cases and severe infections caused by the severe acute respiratory syndrome coronavirus 2 (SARS-CoV-2) has started across the world [1]. It was referred to as the fourth COVID-19 wave. Among the main causes for this trend were the emergence of the delta variant as the leading variant of new cases worldwide and the possible waning of vaccine-elicited immunity [2–5]. The reality in Israel was not different, with a similar rise in confirmed cases, reaching a peak of 2300 cases per million per day in early September 2021 [1]. During the previous rise of Covid-19 (the third wave, December 2020 to March 2021), the Israeli Ministry of Health (MoH) launched an early and mass vaccination campaign. Starting on mid-December 2020 over half of the adult population received two doses of vaccine (mainly BNT162b2) during a time frame of 3 months. Vaccines proved to be highly effective, with a decline in incidence of new confirmed cases to minimum by end of April 2021 [6–8]. Due to the mentioned above a booster vaccine was approved in Israel after the beginning of the fourth Covid-19 wave. Booster administration was reinforced by studies showing a decline in vaccine efficacy and lower levels of neutralizing antibodies [3, 5, 6, 9]. Booster vaccination in Israel was initiated in mid-July 2021, reaching over 40% of the population by November 2021. The booster administration was shown to reduce new confirmed or severe Covid-19 cases in studies among over one million persons [10]. Real life data showed similar results, as only 64 new confirmed cases per day per million occurred by the end of October 2021. The fourth wave of Covid-19 in Israel presented new challenges for medical providers. Unlike before, the elapsed time from the first two vaccine doses and the new booster created a variety of vaccination status among patients, changing throughout the wave and possibly effecting their disease presentation and progression. The aim of this single-center cohort study is to describe and examine the associations between severe COVID-19 patients' vaccine status and their characteristics and disease outcomes during the fourth COVID-19 wave.

## Materials and methods

### 2.1 Study design and participants

We conducted a retrospective observational study of all consecutive patients admitted due to severe COVID-19 in a large tertiary center through a three-month period during the fourth wave (between July 25 and October 25, 2021). All patients included in the study had a positive nasopharyngeal real-time polymerase chain reaction (PCR) result for SARS-CoV-2 and a severe or critical illness definition during their hospital stay. We obtained data regarding the demographics, medical history, COVID-19 immune status, clinical information, and laboratory results from the medical electronic records. The study was approved by the Sourasky Medical Center review board (TLV-0876-20). Informed consent was waived for this study.

We chose to evaluate patient characteristics and clinical factors based on previous literature [11–14]. Baseline characteristics included age (as continuous variable), gender and comorbidities such as diabetes mellitus (DM), hypertension (HTN), obesity (defined as BMI over 30 kg/

m$^2$), heart failure (HF), chronic kidney disease (CKD), cognitive decline, etc. Evaluated laboratory tests were the highest test value that was recorded during admission and included C-reactive protein (CRP), lactate dehydrogenase (LDH), creatine phosphor kinase (CPK), d-dimer and troponin. Disease outcomes included hospital stay duration (days), admission to intensive care unit (ICU), level of oxygen support needed, mechanical ventilation (intubation), extra-pulmonary complications and secondary infections. We tested antibody levels using the Siemens' semiquantitative receptor-binding domain (RBD) IgG assay, authorized by the U.S. Food and Drug Administration [15]. SARS-CoV-2 IgG (sCOVG) index value lower than 1.0 was considered negative, per manufacturer instructions. We stratified the sCOVG result as high if the index value was 10 or greater, based on data showing this value to correspond with pseudo-virus neutralization titers and its use in previous studies [16, 17]. For antibody analysis we included only patients who were vaccinated twice with antibody levels taken earlier than 10 days before first positive PCR test.

## 2.2 Definitions

Illness severity was defined based on the National Institutes of Health (NIH) and Israel Ministry of Health (MoH) definitions [18, 19]. The main definitions for severe illness were oxygen saturation ≤93% while breathing ambient air at resting state or respiratory rate ≥30/min. Critical COVID-19 disease included any of the following: requirement of invasive ventilation, non-invasive ventilation via continuous positive airway pressure (CPAP) or high-flow nasal cannula (HFNC), presence of shock or severe organ failure requiring ICU care. Oxygen supplement was given to achieve an oxygen saturation above 94% at rest, using nasal cannula, low flow mask (venturi mask) or reservoir mask, by their supplement capacity from lower to highest respectively. COVID-19 vaccine status was divided to patients defined as non-vaccinated, patients after the first two doses of the BNT162b2 vaccine and patients after the booster dose. Patients were considered vaccinated if the last vaccine dose was given at least 10 days before the first positive nasopharyngeal PCR test for SARS-CoV-2, as previous studies shown a protective effect starting 7 to 12 days after vaccinations [10, 20, 21]. Only two patients received the first vaccine dose without the second dose, both more than six months before their hospital arrival. For this reason, both patients were included in the non-vaccinated group. Only one patient had a prior COVID-19 infection and was added to a group based on his vaccine status only. Immune deficiency state included active hematologic malignancy or treatment with immunosuppressive drugs (steroids were considered immunosuppression if administered for more than one month in an equivalent dose of 15 mg prednisone). Secondary infections were defined by symptoms not attributed to COVID-19 with a relevant positive culture or serology.

## 2.3 Treatment

We followed the institute treatment guidelines as there are no official guidelines by Israel MoH. These guidelines are in accordance with previous studies and the NIH guidelines [22–24]. Our treatment regimen for severe COVID-19 patients included dexamethasone (6 mg every 24 h, for up to 10 days) and a prophylaxis daily dose of enoxaparin. Remdesivir was given per case based on infectious disease expert opinion (200 mg loading and 100 mg per day for 4 days). Patients that presented with a deteriorating respiratory distress characterized by rapidly increasing oxygen demands, received tocilizumab (8 mg/kg in one or two doses). On august 19th, 2021 a shortage with tocilizumab occurred which caused its substitution with baricitinib. Patients presenting with moderate or early severe disease or those with immunodeficiency and low sCOVG antibody levels received the anti-SARS-CoV-2 antibodies casirivimab plus imdevimab (REGEN-COV) as per local guidelines.

## 2.4 Statistical analysis

Data were analyzed with IBM SPSS statistics software version 28.0. The significance levels were set at 0.05. Data were presented as mean and standard deviation for continuous variables and as frequency and percentage for categorical variables. Chi-Square tests and ANOVA tests were used for differences between the 3 groups (non-vaccinated, received the first two doses, received the booster dose). We used the multivariate Ordinal Regression to correlate the dependence of the ordinal vaccinate status with a set of predictors. Chi-Square tests and independent t- tests were used for differences between two groups. The logistic regression model was used to predict death/ICU based on the variables that were significant in the univariate analysis.

## Results

### 3.1 Trends in admissions by COVID-19 immune status

During the study period, 436 patients were admitted to our center with active COVID-19. Of all admitted patients, 349 had severe or critical disease and were included in the study. Fig 1 shows the number of patients admitted to our center due to severe COVID-19 over the study period divided by their vaccine status. At the beginning of the fourth wave patients after the first two vaccine doses were the majority of admitted patients. Non vaccinated patients became the majority of admitted patients by mid-end of august soon after the initiation of the booster vaccinations. Admission of patients that received the booster vaccine started from mid-august and there was no change with this fraction of patients along this study period. As presented in Table 1, nearly 58% of the study cohort were not vaccinated, 35% had the first two vaccine doses and only 7% received the booster vaccine. A sub analysis of admitted patients under the age of 50 shows that only 19% of them had the first two vaccine doses and that none had received a booster (Table 1).

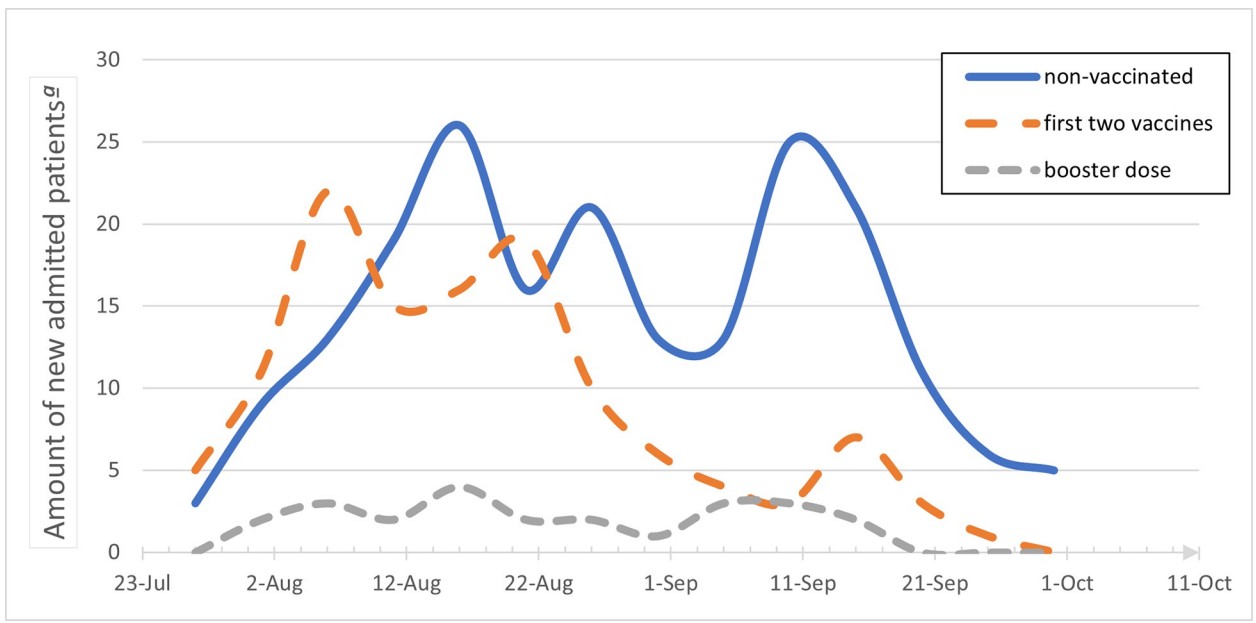

**Fig 1. Trends in hospital admission of severe COVID-19 patients divided by the vaccine status.** [a] Present the total amount of patients admitted during the study period, analyzed by the combined number in 5-days intervals.

**Table 1. Baseline characteristics of hospitalized severe or critical COVID-19 patients by vaccine status.**

| | Total n = 349 | No vaccine n = 202 | Two Vaccines n = 122 | Booster n = 25 | P |
|---|---|---|---|---|---|
| | No. (%) | No. (%) | No. (%) | No. (%) | |
| Age, mean ± SD, y | 67.7± 17.1 | 64± 17 | 72± 16 | 78 ±10 | <0.001 |
| Age ≤ 50 y | 64 (18) | 52 (26) | 12 (10) | 0 | |
| Age > 50 y | 285 (82) | 150 (74) | 110 (90) | 25 (100) | <0.001 |
| Male gender | 200 (57) | 103 (51) | 82 (67) | 15(60) | 0.016 |
| HTN | 186 (53) | 99 (49) | 70 (57) | 17 (68) | 0.107 |
| Hyperlipidemia | 145 (41.5) | 73 (36) | 57 (47) | 15 (60) | 0.026 |
| DM | 112 (32) | 55 (27) | 44 (36) | 13 (52) | 0.022 |
| Obesity (BMI≥30) | 106 (30) | 59 (29) | 43 (35) | 4 (16) | 0.139 |
| COPD | 51 (15) | 30 (15) | 18 (15) | 3 (12) | 0.929 |
| IHD | 68 (19.5) | 24 (12) | 35 (29) | 9 (36) | <0.001 |
| HF | 57 (16) | 25 (12) | 23 (19) | 9 (36) | 0.007 |
| AF | 47 (13.5) | 23 (11) | 17 (14) | 7 (28) | 0.070 |
| Immuno-deficiency [a] | 40 (11.5) | 14 (7) | 19 (16) | 7 (28) | 0.002 |
| cirrhosis | 3 (0.9) | 2 | 1 | 0 | 0.878 |
| CKD | 44 (12.6) | 14 (7) | 26 (21) | 4 (16) | <0.001 |
| Cognitive decline | 70 (20) | 31 (15) | 30 (25) | 9 (36) | 0.016 |
| number of total comorbidities | 2.5 ±2.0 | 2.1± 1.9 | 3.0 ±2.0 | 3.5± 2.4 | <0.001 |
| Symptoms to hosp. [b] | 6.45± 4.36 | 7.2± 4.4 | 5.4± 4.1 | 5.6 ±4.6 | 0.002 |
| PCR to hosp. [b] | 4.56± 4.69 | 4.7± 4.5 | 4.3 ±4.4 | 4.6± 6.8 | 0.792 |

Abbreviations: y, year; HTN, hypertension; DM, diabetes mellitus; BMI, body mass index; COPD, Chronic Obstructive Pulmonary Disease; IHD, ischemic heart disease; HF, heart failure; AF, atrial fibrillation; CKD, chronic kidney disease.

[a] Patients included are those with Immunodeficiency secondary to immuno-suppressive therapy or lymphoproliferative malignancy.

[b] Mean number of days from symptoms or from first positive SARS-COV-2 PCR test to hospital admission.

## 3.2 General characteristics and comparison by vaccine group

Table 1 describes the baseline characteristics of the study cohort. The mean age was 67.7 ± 17.1, 43% were females and the mean number of related comorbidities was 2.5 ± 2.0 per patient. The most prevalent comorbidities were HTN (53%), DM (32%), hyperlipidemia (41.5%) and obesity (30%) (BMI above 30). Forty patients (11.5%) were immunosuppressed, highest among the booster vaccine group (28%). Duration from first symptoms to admission and from positive PCR result to admission were 6.45 ± 4.36 and 4.56 ± 4.69 days, respectively.

Older age (p < 0.001), male gender (p = 0.016), number of comorbidities (p < 0.001), DM (p = 0.022), hyperlipidemia (p = 0.026), ischemic heart disease (p < 0.001), heart failure (p = 0.007), immunodeficient state (p = 0.002), CKD (p < 0.001) and cognitive decline (p = 0.016) were significantly different between the groups, with higher rates among vaccinated patients. All mentioned variables but CKD and gender had a direct relationship with the number of vaccines given (none, first two vaccine doses and booster dose). Using the multivariate regression model (described in S1 Table), age, male gender, and number of comorbidities were independently associated with vaccinated patients.

Treatment during hospitalization is presented in S2 Table. Dexamethasone was given to 95% of the patients. Tocilizumab and baricitinib were given mostly for critical COVID-19 disease and were used in 9.2% and 24% of the patients, respectively. Treatment with tocilizumab, which was available only during the three weeks of the study period (see methods), was more prevalent among patients after the first two vaccines (15%, p = 0.016). Treatment with

**Table 2. Outcomes of hospitalized severe or critical COVID-19 patients by vaccine status.**

| | Total n = 349 | No vaccine n = 202 | Two Vaccines n = 122 | Booster n = 25 | P |
|---|---|---|---|---|---|
| | No. (%) | No. (%) | No. (%) | No. (%) | |
| **COVID-19 worse severity:** [a] | | | | | |
| Severe | 243 (70) | 131 (65) | 90 (74) | 22 (88) | 0.028 |
| Critical | 106 (30) | 71 (35) | 32 (26) | 3 (12) | |
| **Maximal oxygen support needed:** [b] | | | | | |
| Nasal cannula | 179 (52) | 91 (45) | 70 (58) | 18 (72) | 0.004 |
| Oxygen mask | 64 (18) | 40 (20) | 20 (16.5) | 4 (16) | |
| Non-invasive | 57 (16) | 39 (19) | 17 (14) | 1 (4) | |
| Intubation | 47 (13.5) | 31 (15) | 14 (12) | 2 (8) | |
| Death | 70 (20) | 40 (20) | 26 (21) | 4 (16) | 0.825 |
| ICU admission | 36 (10) | 29 (14) | 7 (6) | 0 | 0.010 |
| Hospital duration, mean, days | 8.05± 8.47 | 8.7± 9.5 | 7.2± 6.9 | 6.4± 5.2 | 0.185 |
| Extra–pulmonary [c] | 37 (11) | 18 (9) | 16 (13) | 3 (12) | 0.479 |
| AKI [d] | 88 (25) | 50 (25) | 32 (26) | 6 (24) | 0.947 |
| Other proven infection [e] | 54 (15.5) | 25 (12) | 24 (20) | 5 (20) | 0.172 |

Abbreviations: COVID-19, coronavirus disease 2019; ICU, intensive care unit; AKI, acute kidney injury.

[a] Severity of illness was determined according to the National Institutes of Health and Israel Ministry of Health definitions.

[b] Oxygen mask include the need of a low flow mask (venturi mask) or a reservoir mask; Non-invasive include the need of continuous positive airway pressure (CPAP) or high-flow nasal cannula (HFNC).

[c] Cardiovascular, neurological, and hematological complications attributed to COVID-19 effect.

[d] Acute kidney injury diagnosed as an increase in serum creatinine by $\geq$0.3 milligrams per deciliter (mg/dl) within 48 hours or an increase in serum creatinine to $\geq$1.5 times baseline.

[e] Non COVID-19 infection is defined by symptoms not attributed to covid-19 with a relevant positive culture.

baricitinib, which substituted tocilizumab, was given to 31% of the non-vaccinated group compared with 14% and 16% in the two-vaccines and booster vaccine groups, respectively (p = 0.002).

## 3.3 Disease outcomes and comparison by vaccine group

Disease outcomes are presented in Table 2. Overall, 106 (30%) patients had a deterioration to critical disease, 47 (13.5%) required intubation, 36 (10%) were admitted to the ICU and 70 (20%) died in the hospital. The maximal oxygen support needed was different between the groups (p = 0.004), with higher support level among non-vaccinated patients. Similar trend was found for critical COVID-19 disease and ICU admissions (p < 0.05 and p = 0.01, respectively). Mean hospital duration was 8 ± 8.5 days and was not different between the groups (p = 0.185). Acute kidney injury occurred in 25% of the patients and other extra-pulmonary complications, including neurologic, hematologic, and cardiovascular occurred in 11.2%. Fifty-four patients (15.5%) had additional non-COVID-19 infection, most commonly due to respiratory (26, 48%) or urinary tract (17, 31%) infections. Of the evaluated laboratory results (S3 Table), only maximal LDH was different between the groups (p < 0.005), highest among the non-vaccinated patients (1012 ± 1211 U/L).

Analysis of risk factors for the combined outcome of death or admission to the ICU is presented in S4 Table. Variables found to be associated with the combined outcome were HTN (p = 0.017), BMI (p = 0.021), heart failure (p = 0.023) and higher total number of comorbidities (p < 0.001). Higher levels of D-dimer, CPK, LDH and CRP were also significantly associated with this combined outcome. In a multivariant analysis including selected patient

characteristics and clinical factors (S5 Table), only BMI was an independent predictor for death or ICU admission (HR 1.7, p = 0.048).

### 3.4 Characteristics by antibody levels

Seventy-one patients of the 122 who received the first two vaccine doses were included in the antibody cohort, based on the inclusion criteria described above. S6 Table presents baseline patient characteristics divided by low/high levels of sCOVG antibodies (index level under/ above 10 respectively). Only older age was found to be significantly associated with low antibody level (p < 0.001). We found that extra-pulmonary COVID-19 complications and secondary non-COVID-19 infections were significantly associated with lower sCOVG level. Disease outcomes for both low/high sCOVG levels and negative/positive sCOVG levels (index level under/above 1 respectively) are presented in Table 3. Rates of critical disease, need for intubation, admission to ICU, death and hospital duration of stay were not associated with antibody levels.

## Discussion

Severe COVID-19 disease usually requires hospitalization (mainly for oxygen support), as opposed to mild or moderate disease. Increasing number of COVID-19 related hospitalizations lead to shifting of medical staff towards designated COVID-19 units and reaching towards the edge of hypothetical insufficiency of the health care system. Severe patients can rapidly deteriorate, requiring even more resources and manpower. We believe that severe level of disease causes a major burden on the patients, medical staff, and the health-care system. Vaccines are now the main tool for disease prevention, leading us to focus on its association specifically with severe COVID-19 disease.

The early phase of the fourth COVID-19 wave was characterized by hospitalized patients that received the first two vaccine doses, as presented in Fig 1. A recent Israeli study also showed that during July 2021, two thirds of the cases of severe COVID-19 in Israel occurred in

**Table 3. Disease outcomes by SARS-CoV-2 IgG antibody levels among the antibody cohort.**

| | Low / High sCOVG [a] | | | Negative / Positive sCOVG [a] | | |
|---|---|---|---|---|---|---|
| | Index <10 N = 31 (%) | Index >10 N = 40 (%) | P | Index <1 N = 15 (%) | Index >1 N = 56 (%) | P |
| **COVID-19 worse severity:** [b] | | | | | | |
| Severe | 23 (74) | 30 (75) | 0.938 | 10 (67) | 43 (77) | 0.424 |
| Critical | 8 (26) | 10 (25) | | 5 (33) | 13 (23) | |
| **Maximal oxygen support needed:** [c] | | | | | | |
| Nasal cannula | 18 (60) | 23 (58) | 0.888 | 8 (53) | 33 (60) | 0.554 |
| Oxygen mask | 4 (13) | 8 (20) | | 2 (13) | 10 (18) | |
| Non-invasive | 4 (13) | 4 (10) | | 3 (20) | 5 (9) | |
| Intubation | 4 (13) | 5 (13) | | 2 (13) | 7 (13) | |
| Death | 6 (19) | 8 (20) | 0.946 | 4 (27) | 10 (18) | 0.446 |
| ICU admission | 1 (3) | 4 (10) | 0.269 | 0 | 5 (9) | 0.23 |
| Hospital duration, d | 9.1± 8.3 | 7.1 ±7.4 | 0.298 | 9.7± 9.0 | 7.5± 7.5 | 0.345 |

Abbreviations: sCOVG, SARS-CoV-2 IgG; COVID-19, coronavirus disease 2019; ICU, intensive care unit.

[a] Antibody cohort included only admitted patients after the first two vaccine doses with levels taken earlier than 10 days before first positive PCR swab.

[b] Severity of illness was determined according to the National Institutes of Health and Israel Ministry of Health definitions.

[c] Oxygen mask include the need of a low flow mask (venturi mask) or a reservoir mask; Non-invasive include the need of continuous positive airway pressure (CPAP) or high-flow nasal cannula (HFNC).

persons who received two doses of the BNT162b2 vaccine [6]. Between December 2020 and March 2021, almost 75% of the first two vaccine doses were administered, more than 4 months before the fourth COVID-19 wave [1]. The time gap from the first two doses of vaccines can be the reason for the hospitalization rate due to a waning immune effect [3, 6, 25]. Vaccine's effect was shown to be reduced by a factor of four after 4–7 months [5]. While Chemaitelly et al. [26] showed similar findings, they found that the effectiveness of vaccination against hospitalization and death did not decline after 6 months. Another explanation for the relative high admission number in Israel is the large proportion of vaccinated people among the older adult population. The second part of the fourth wave started from mid-august 2021 and was characterized mainly by admission of non-vaccinated patients. Booster vaccine was available in Israel for people over the age of 60 on August 1st, 2021, and for all adults from early September 2021. By mid-September 2021, over 30% of the population in Israel received the booster dose [27]. This trend highlights the efficacy of the booster vaccine in preventing severe disease, as shown in recent studies [10, 28].

Among the non-vaccinated patients in our study, main baseline characteristics were in accordance with studies from previous COVID-19 waves [12, 29–32]. Hospitalized vaccinated patients were older, predominantly males and had more comorbidities. These associations were in a "dose response" relationship for most variables, meaning a higher incidence for patients that received more vaccines. Cohorts of hospitalized non-vaccinated COVID-19 patients also showed a younger age (52–72 years) and lower rates of comorbidities such as DM (21%-32%), HTN (50%-57%) and cognitive decline (16%), compared with vaccinated patients in our cohort [12, 29, 31–35]. We found a shorter time between first COVID-19 symptoms and admission among vaccinated patients. It is possible that because of their baseline characteristics, vaccinated patients had higher likelihood of comorbidity-related complications secondary to COVID-19 and were less reluctant to arrive to the hospital. The general mean hospital duration was 8 days, which is shorter than several previous cohorts (10–13 days) [11, 32, 33]. Only severe patients were included in our cohort, further highlighting the gap in hospital duration from previous studies. Better treatment options, experience gained from previous COVID-19 waves and established community infrastructure seem to be the reasons for this gap.

Severe COVID-19 disease has high rates of adverse outcomes. Over 15% of hospitalized COVID-19 patients need intubation and 20% die [30, 32, 35, 36]. Our study showed similar findings regarding intubation and death. We found higher rate of critical disease (30%) and lower rate of ICU admission (10%) than previously described [7, 31, 32, 37]. Risk factors for admission to ICU or death in our study are HTN, obesity, HF, and total number of comorbidities. Different risk factors for adverse outcomes among severe COVID-19 patients have been described in the literature. They include older age, male gender, DM, HTN, chronic cardiac disease, obesity, cognitive decline, and CKD [11, 12, 29, 30, 34]. In our study all the reported risk factors were more prevalent among the vaccinated patients, yet these patients were less likely to develop critical disease, used lower levels of oxygen support and had less ICU admissions. Vaccination was the only protective factor among these patients, possibly leveling or even improving their outcomes compared with younger and healthier patients. Hospital stay duration was similar between the older comorbid vaccinated patients compared to the non-vaccinated patients (p = 0.185). Once again, the importance of the vaccines, especially for older patients with comorbidities is highlighted.

It is important to note that the incidence of death was not different between the groups (p = 0.825), despite the significantly higher rates of risk factors among vaccinated patients. A similar finding was found in a study of 152 vaccinated patients hospitalized with COVID-19 infection [36]. Other reasons such as patient comorbidities or the baseline condition of the

patient may have contributed to their admission outcome. Zooming in towards the booster group, 4 patients had died (16%) with a mean of 5 comorbidities, three of them with a cognitive decline. Physicians in our center emphasize the need to determine long term goals, including do not resuscitate orders and advanced planning of care. It should be considered for effecting patient outcomes and therefore the study results.

Until the end of October 2021, the center for disease control and prevention (CDC) stated that a booster shot for patients under the age of 50, may be given only if an underlying medical condition exists [38]. During the study period, no patient under the age of 50 was admitted after receiving the booster dose. This contrasts with the fact that more than 50% of the adult people under the age of 50 received the booster dose by November 2021 [27]. There was a significantly higher proportion of patients under 50 in the non-vaccinated group compared to patients after the first two vaccines (26% vs 10%, p < 0.001). Like the general study population, there was no significant change in death or ICU admission between the groups (18% vs 24%, p = 0.48). The number of patients in these groups is small, but still strengthen the current recommendation to vaccinate all adults above the age of 18.

The prognostic value of antibody levels for COVID-19 among vaccinated people is still debated. Antibody levels were shown to correlate mainly with COVID-19 infection risk but not specifically with severe infection [9]. Brosh-Nissimov et al, showed that while hospitalized vaccinated patients with poor outcomes had lower antibody levels at admission, it was not a significant association [36]. Low or high levels of sCOVG (index level of 10 as cutoff) was not associated with level of oxygen support, ICU admission or death. Prior study demonstrated that low levels of antibodies are needed to protect from severe disease compared with protection from any COVID-19 infection [39]. Comparison by negative or positive sCOVG (index level of 1 as cutoff) also showed no association with the outcomes. As expected Casirivimab plus imdevimab were given in significantly higher rates to patients with low antibody levels (42% vs 8%, p < .001) and may have affected patient outcomes. We found older age to be a significant risk factor for low sCOVG level, as presented in previous studies [40, 41]. An interesting finding was the higher rate of extra-pulmonary COVID-19 complications and other secondary infections in patients with lower sCOVG levels. Larger controlled studies are needed to further evaluate this association.

The study has some limitations. The retrospective nature of our study prevented us from estimating the risk factors for vaccine failure, because patients were identified after hospitalization and were not compared with vaccinated uninfected controls. Although the study population showed similar characteristics with previous studies from Israel and world-wide, generalizability of these results should consider the single-center nature of this study. Other disease outcomes from additional sub-analysis of patients after the booster dose were limited by their small numbers. Factors like complications secondary to a patients' comorbidity or a "do not intubate" order by the patient could have affected major outcomes, specifically intubation and death, but were not assessed in the study. We also did not have access to readmissions in other medical centers or to ambulatory medical follow-up. Our findings concerning the sCOVG antibody levels do not necessarily represent the levels achieved by the first two vaccine dose. By only addressing time from first positive SARS-CoV-2 PCR for inclusion to the antibody cohort, the antibody levels may also be affected by the present COVID-19 infection.

## Conclusions

This study describes the association of vaccine status with characteristics and outcomes of patients hospitalized with severe COVID-19 during the fourth wave in Israel. While hospitalized vaccinated patients had a significantly higher rates of most known risk factors for

COVID-19 adverse outcomes, they needed a lower level of oxygen support and had lower rates of critical disease or ICU admissions. These findings had a direct relationship to the number of vaccines given. Death and hospital stay duration were similar between the vaccine status groups. Our results emphasize the positive effects of the booster vaccine, especially for older comorbid patients.

## Supporting information

**S1 Table. Multivariate regression for correlation of selected variables with vaccine status.**
(DOCX)

**S2 Table. Treatment during hospitalization and correlation to vaccine status.**
(DOCX)

**S3 Table. Maximal values of Selected lab results during hospitalization and correlation to vaccine status.**
(DOCX)

**S4 Table. Characteristics of patients by the combined outcome of ICU admission or death.**
(DOCX)

**S5 Table. Multivariate analysis of significant risk factors for death or ICU admission.**
(DOCX)

**S6 Table. Baseline characteristics and outcomes by SARS-CoV-2 IgG antibody levels among the antibody cohort.**
(DOCX)

## Author Contributions

**Conceptualization:** Ophir Freund, Luba Tau, Gil Bornstein.

**Formal analysis:** Ophir Freund, Tali Epstein Weiss, Lior Zornitzki, Shir Frydman.

**Funding acquisition:** Gil Bornstein.

**Investigation:** Luba Tau, Tali Epstein Weiss, Giris Jacob.

**Methodology:** Luba Tau.

**Project administration:** Gil Bornstein.

**Visualization:** Lior Zornitzki, Shir Frydman.

**Writing – original draft:** Ophir Freund.

**Writing – review & editing:** Luba Tau, Tali Epstein Weiss, Lior Zornitzki, Shir Frydman, Giris Jacob, Gil Bornstein.

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
