## [Decision Letter · Decision Letter 0]

24 Feb 2022

PONE-D-22-00969Associations of vaccine status with characteristics and outcomes of hospitalized severe COVID-19 patients in the booster era.PLOS ONE

Dear Dr. Freund,

Thank you for submitting your manuscript to PLOS ONE. After careful consideration, we feel that it has merit but does not fully meet PLOS ONE’s publication criteria as it currently stands. Therefore, we invite you to submit a revised version of the manuscript that addresses the points raised during the review process.

We look forward to receiving your revised manuscript.

Kind regards,

Linglin Xie

Academic Editor

PLOS ONE

Journal Requirements:

Reviewers' comments:

Reviewer's Responses to Questions

**Comments to the Author**

1. Is the manuscript technically sound, and do the data support the conclusions?

Reviewer #1: Yes

Reviewer #2: Partly

2. Has the statistical analysis been performed appropriately and rigorously? 

Reviewer #1: No

Reviewer #2: Yes

3. Have the authors made all data underlying the findings in their manuscript fully available?

Reviewer #1: Yes

Reviewer #2: No

4. Is the manuscript presented in an intelligible fashion and written in standard English?

Reviewer #1: Yes

Reviewer #2: Yes

5. Review Comments to the Author

Reviewer #1: This article entitled with “Associations of vaccine status with characteristics and outcomes of hospitalized severe COVID-19 patients in the booster era” compared the characteristics and outcomes in hospitalized severe covid-19 patients among non-vaccine, two vaccines and booster group, which could be an important guidance to better fight COVID-19 together with other clinical studies. As the author mentioned, there were several similar studies published, so this article is a bit lacking in novelty. In addition, I have some minor comments listed below:

1. Only male gender was included in the study, then sexual difference should be discussed.

2. What does the P value stand for in Table 1, 2, S1 and S2, since there were three different groups divided by no vaccine, two vaccines and booster?

Reviewer #2: In this manuscript, Freund et al. identified and summarized characteristics of outcomes of COVID-19 patients according to their vaccine status (grouped into three) during the fourth wave in Israel. The association of vaccine status with outcomes of hospitalized patients is well put and studied. However, I have some concern about the design of study, especially the categorization of patient groups, which also leads to the obvious limitation of the study due to relatively low number of patient samples from the booster group, although the statistical methods are all properly implemented.

In addition, there are some big grammatical and formatting issues throughout the manuscript. Please make significant efforts to correct and proofread it before re-submission. Comments are listed below.

Major

--What biases/differences would it present to the study if the authors single out patients that were vaccinated only once? (i.e., first dose only, without the second dose). This would lead to four groups: Nonvaccinated (zero dose) vs. one dose vs. two doses (fully vaccinated) vs. booster dose. I believe it would strengthen the design of this study.

--“Tocilizumab and baricitinib were given mostly for critical COVID-19 disease and were used in 9.2% and 24% of the patients, respectively”. This is helpful, but what is the exact ratio/percentage of the usage of these two in the three different groups, respectively?

--There is no clear explanation or descriptive summary based on the results of Table 1.

-- Higher level of oxygen support was inversely proportional to the number of vaccines given (p = 0.004). Is this referring to Table 2? Please cite properly. Also, this s entence is a bit confusing. When you state two variables to be inversely proportional, avoid using comparative forms.

--Please cite the tables in the correct order as in which they appear in the manuscript. Citations in the results sections seem very disorganized. It seems messed up in the results section. Also, where would you cite Table S2? In addition, Table S2 row 3 is poorly formatted.

--Given the limitation of the study, especially the small number of patients after the booster shoot, I do not find strong implications for this design. The ratio between the two groups (no vaccine vs two vaccines) is acceptable, and it could even be improved by taking the suggestion from my first comment. However, such a low patient number for booster group seems like a concern to me.

--In addition to my last comment, does any of the group include patients who got reinfection?

Minor

Grammatical: “clinical information and laboratory results were obtained by reviewing each patient medical electronic records”. Please rephrase correctly.

Grammatical: “Patients were defined as critically ill if met any of the following criteria:” Please rephrase.

Formatting: “Patients presented with deteriorating respiratory distress e.g., rapidly increasing oxygen demands”, format the use of e.g., properly.

Grammatical: “During the study period 436 patients were admitted to our center with active COVID-19, of them 349 met the definition of severe or critical disease and were included in the study.” Please rephrase.

Please include table caption above the table instead of placing it underneath.

6. PLOS authors have the option to publish the peer review history of their article (what does this mean?). If published, this will include your full peer review and any attached files.

Reviewer #1: No

Reviewer #2: No

---

## [Author Response · Author response to Decision Letter 0]

24 Mar 2022

Response to all reviewers and editor comments are in the "Response to Reviewers" file attached.

---

## [Decision Letter · Decision Letter 1]

21 Apr 2022

Associations of vaccine status with characteristics and outcomes of hospitalized severe COVID-19 patients in the booster era.

PONE-D-22-00969R1

Dear Dr. Freund,

We’re pleased to inform you that your manuscript has been judged scientifically suitable for publication and will be formally accepted for publication once it meets all outstanding technical requirements.

Kind regards,

Linglin Xie

Academic Editor

PLOS ONE

Additional Editor Comments (optional):

Reviewers' comments:

Reviewer's Responses to Questions

**Comments to the Author**

1. If the authors have adequately addressed your comments raised in a previous round of review and you feel that this manuscript is now acceptable for publication, you may indicate that here to bypass the “Comments to the Author” section, enter your conflict of interest statement in the “Confidential to Editor” section, and submit your "Accept" recommendation.

Reviewer #1: All comments have been addressed

Reviewer #2: All comments have been addressed

2. Is the manuscript technically sound, and do the data support the conclusions?

Reviewer #1: Yes

Reviewer #2: Yes

3. Has the statistical analysis been performed appropriately and rigorously? 

Reviewer #1: Yes

Reviewer #2: Yes

4. Have the authors made all data underlying the findings in their manuscript fully available?

Reviewer #1: Yes

Reviewer #2: No

5. Is the manuscript presented in an intelligible fashion and written in standard English?

Reviewer #1: Yes

Reviewer #2: Yes

6. Review Comments to the Author

Reviewer #1: (No Response)

Reviewer #2: The authors have addressed my comments properly.

I still want to propose some minor formatting issue (which was one of my comments). Please format the "mean +/- SD" in each table and be consistent. There is one in Table S3, S4, and 5 in Table S6 that need correction. Also, pay attention to those inconsistencies in the main Tables as well. (Tables 1-3). Thanks.

7. PLOS authors have the option to publish the peer review history of their article (what does this mean?). If published, this will include your full peer review and any attached files.

Reviewer #1: No

Reviewer #2: No

---

## [Editor Report · Acceptance letter]

29 Apr 2022

PONE-D-22-00969R1 

Associations of vaccine status with characteristics and outcomes of hospitalized severe COVID-19 patients in the booster era. 

Dear Dr. Freund:

I'm pleased to inform you that your manuscript has been deemed suitable for publication in PLOS ONE. Congratulations! Your manuscript is now with our production department. 

Kind regards, 

on behalf of

Dr. Linglin Xie 

Academic Editor

PLOS ONE